

# *Clostridioides difficile* infection after cardiac surgery: Assessment of prevalence, risk factors and clinical outcomes—retrospective study

Anna Rzucidło-Hymczak[1], Hubert Hymczak[2], Aldona Olechowska-Jarząb[3], Anna Gorczyca[1], Boguslaw Kapelak[4], Rafał Drwiła[5] and Dariusz Plicner[6]

[1] Department of Infectious Diseases, John Paul II Hospital, Krakow, Poland
[2] Department of Anesthesiology, John Paul II Hospital, Krakow, Poland
[3] Department of Microbiology, John Paul II Hospital, Krakow, Poland
[4] Department of Cardiovascular Surgery and Transplantology, John Paul II Hospital, Krakow, Poland
[5] Medical College, Jagiellonian University, Krakow, Poland
[6] Unit of Experimental Cardiology and Cardiac Surgery, Faculty of Medicine and Health Sciences, Andrzej Frycz Modrzewski Krakow University, Krakow, Poland

Corresponding author
Dariusz Plicner, dplicner@afm.edu.pl

## ABSTRACT

**Background.** Clostridioides difficile infection (CDI) is the most common cause of hospital-acquired diarrhea. There is little available data regarding risk factors of CDI for patients who undergo cardiac surgery. The study evaluated the course of CDI in patients after cardiac surgery.

**Methods.** Of 6,198 patients studied, 70 (1.1%) developed CDI. The control group consisted of 73 patients in whom CDI was excluded. Perioperative data and clinical outcomes were analyzed.

**Results.** Patients with CDI were significantly older in comparison to the control group (median age 73.0 vs 67.0, $P = 0.005$) and more frequently received proton pump inhibitors, statins, $\beta$-blockers and acetylsalicylic acid before surgery ($P = 0.008, P = 0.012, P = 0.004$, and $P = 0.001$, respectively). In addition, the presence of atherosclerosis, coronary disease and history of malignant neoplasms correlated positively with the development of CDI ($P = 0.012, P = 0.036$ and $P = 0.05$, respectively). There were no differences in the type or timing of surgery, aortic cross-clamp and cardiopulmonary bypass time, volume of postoperative drainage and administration of blood products between the studied groups. Relapse was more common among overweight patients with high postoperative plasma glucose or patients with higher C-reactive protein during the first episode of CDI, as well as those with a history of coronary disease or diabetes mellitus ($P = 0.005$, $P = 0.030, P = 0.009, P = 0.049$, and $P = 0.025$, respectively). Fifteen patients died (21.4%) from the CDI group and 7 (9.6%) from the control group ($P = 0.050$). Emergent procedures, prolonged stay in the intensive care unit, longer mechanical ventilation and high white blood cell count during the diarrhea were associated with higher mortality among patients with CDI ($P = 0.05, P = 0.041, P = 0.004$ and $P = 0.007$, respectively).

**Conclusions.** The study did not reveal any specific cardiac surgery-related risk factors for development of CDI.

## INTRODUCTION

*Clostridioides difficile* (CD) is widely spread in the human environment and present in about 7–18% of the adult population (*Donskey, Kundrapu & Deshpande, 2015*). CD infection (CDI) is the most common cause of hospital-acquired diarrhea and may follow a severe course with many complications, which can include fatal colitis (*Cunha, 1998*; *Ricciardi et al., 2007*). Despite increased efforts to prevent this infection, the incidence and severity of nosocomial CDI has continued to grow worldwide (*Ricciardi et al., 2007*; *Clements et al., 2010*).

Known risk factors for CDI include advanced age, female gender, use of broad-spectrum antibiotics, use of proton pump inhibitors (PPI), chronic comorbidities, immunocompromised states and prolonged, multiple hospital stays (*McFarland, Surawicz & Stamm, 1990*; *Eze et al., 2017*; *De Roo & Regenbogen, 2020*; *Furuya-Kanamori et al., 2015a*). Patients who undergo surgery present additional risks for CDI associated with catheter-related infections, prolonged mechanical ventilation, extensive blood product usage, indwelling catheter drainage and open cavities (*Gelijns et al., 2014*).

There is little available data regarding risk factors for CDI among patients who undergo cardiac surgery. There have only been a few reports investigating the risk of CDI in patients after heart procedures (*Gelijns et al., 2014*; *Vondran et al., 2018*; *Flagg et al., 2014*; *Kirkwood et al., 2018*). This prompted us to evaluate the prevalence of hospital-acquired CDI after cardiac surgery, identify patient characteristics and detect risk factors for CDI. Moreover we assessed the course of the disease and final outcomes for this group of patients.

## MATERIALS & METHODS

### Patients

Between January 2014 and December 2016, a total of 6,198 adult patients underwent cardiac surgery in our hospital. Seventy of the patients were diagnosed with CDI. The control group was comprised of 73 patients for whom CDI had been excluded and this group was matched to the group of CDI patients by the date of surgery.

Demographics, comorbidities, type and timing of cardiac surgery, operative characteristics, perioperative antibiotic use, exposure to known risk factors for CDI and in-hospital mortality were collected retrospectively. Additionally, length of hospitalization until the onset of diarrhea, severity and recurrence of the disease, methods of treatment and seasonal distribution of CDI were obtained.

CDI was suspected in each patient who experienced diarrhea (defined as the passage of three or more unformed stools per day). CDI was defined as a combination of symptoms and signs of the disease and confirmed by microbiological evidence of toxin-producing CD in the patients' stools (*Debast, Bauer & Kuijper, 2014*). Stool samples were analyzed using the rapid enzyme immunoassays test, C. Diff Quik Chek Complete test (Techlab, Orlando, USA).

Non-severe CDI was defined by a white blood cell (WBC) count of $\leq$ 15,000 cells/mL and a serum creatinine level $<$ 1.5 mg/dL. Severe CDI was specified by a WBC count of $\geq$ 15,000 cells/mL or a serum creatinine level $>$1.5 mg/dL. Criteria for fulminant CDI included occurrence of hypotension or shock, ileus or megacolon (*McDonald et al., 2018*).

The treatment of CDI was consistent with the 2010 recommendations (*Cohen et al., 2010*). Metronidazole was the drug of choice for an initial episode of non-severe CDI and vancomycin was the drug of choice for an initial episode of severe CDI. Combination therapy with oral or rectal vancomycin and intravenously administered metronidazole was the regimen of choice for the treatment of severe, complicated or fulminant CDI.

Recurrence of the disease was identified as a relapse within 8 weeks after the onset of a previous episode (*Debast, Bauer & Kuijper, 2014*). Stress hyperglycemia was defined as one or more blood sugar measurements $>$ 180 mg/dL during the first postoperative 24 h (*Gelijns et al., 2014*). In-hospital mortality was specified as death occurring during the same hospitalization stay as the cardiac surgery.

Each patient received periprocedural antimicrobial prophylaxis (most often the first generation of cephalosporin). Routine laboratory variables were determined using standard laboratory techniques. Study protocol was approved by the local Research Ethics Committee (Andrzej Frycz Modrzewski Krakow University, Krakow, Poland 10/2019). Verbal consent of patients was acquired.

## Statistical analysis

Descriptive statistics were described as numbers and percentages for categorical variables. Continuous variables were presented as mean ($\pm$ standard deviation) or median and quartiles, as appropriate. Normality was assessed using the Shapiro–Wilk test. Equality of variances was assessed using Levene's test. Differences between groups were compared using the Student's or Welch's $t$-test depending on the equality of variances for normal distributed variables. The Mann–Whitney U test was used for non-normal distributed continuous variables. Nominal variables were compared by the Pearson's chi-square test or Fisher's exact test if 20% of cells had an expected count of less than 5. Significance was accepted at $P \leq 0.05$. Statistical analyses were performed with JMP®, Version 14.2.0 (SAS Institute INC., Cary, NC, USA) and using R, Version 3.4.1 (R Core Team. R: A language and environment for statistical computing. R Foundation for Statistical Computing. Vienna, Austria, 2017. https://www.r-project.org/).

# RESULTS

## Baseline characteristics

Of the 6,198 patients, 70 (1.1%) developed CDI. Patients with CDI were significantly older in comparison to the control group (median age 73.0 vs 67.0, $P = 0.005$). There was no correlation between gender and incidence of CDI ($P = 0.595$). The European System for Cardiac Operative Risk Evaluation (EuroSCORE) values were higher in patients with CDI ($P < 0.001$). Patients with CDI more often received PPI, statins, β-blockers and acetylsalicylic acid before surgery ($P = 0.008$, $P = 0.012$, $P = 0.004$, and $P = 0.001$, respectively). In addition, the presence of atherosclerosis, coronary disease, and history of malignant

neoplasms correlated positively with the development of CDI ($P = 0.012$, $P = 0.036$ and $P = 0.05$, respectively). Patients in the CDI group were hospitalized more often during the six months prior to the surgery ($P = 0.001$). Mean preoperative hospitalization time in the cardiac surgery ward was $1.5 \pm 0.2$ days. Other baseline variables were comparable among groups (Table 1).

## Perioperative characteristics

The most common surgical procedures performed before CDI were valvular heart surgery and coronary artery bypass grafting (41.4% and 35.7%, respectively). There were no differences between the studied groups as far as the type or timing of surgery, aortic cross-clamp and cardiopulmonary bypass time, volume of postoperative drainage, administration of blood products, value of postoperative ejection fraction and frequency of reoperations. Patients with CDI more frequently received additional antibiotics ($P = 0.014$). During the early postoperative course patients with CDI had a significantly higher glucose level and were exposed more frequently to stress hyperglycemia ($P < 0.001$ for both comparisons). During the preoperative period, as well as after surgery, patients with CDI had a significantly lower WBC count ($P = 0.007$ for both comparisons). Other intra- and postoperative variables were similar in both groups (Table 2).

## Course of *Clostridioides difficile* infection

The type of antibiotic therapy used before the first episode of CDI, median times of the disease diagnosis and severity of the infection are shown in Table 3. Five patients in whom fulminant CDI developed, underwent emergency laparotomy and two patients died due to extensive multiple organ failure.

All patients with CDI were treated with oral metronidazole, oral vancomycin or both (intravenous metronidazole with oral vancomycin). Fidaxomicin was not used in our department during the study period (Table 3). Two patients underwent a fecal microbiota transplant during recurrence of the disease.

Although baseline and early postoperative WBC count were significantly lower in patients with CDI, during the course of the disease, WBC count was similar among analyzed groups ($P = 0.139$). In contrast, C-reactive protein (CRP) was higher in the CDI group during this period ($P < 0.001$).

There were no differences in the incidence rate of CDI between the analyzed years although a seasonal pattern was observed. Most cases occurred in March ($n = 14$, 20%) and the least number of cases were found in January ($n = 2$, 3%), ($P = 0.038$ between March and rest of the months) (Fig. 1).

In 9 patients (12.9%) CDI recurred during hospitalization. Mean time of relapse was $27.7 \pm 11.8$ days after the first episode of the disease. Recurrent CDI was more common in overweight patients having high plasma glucose just after surgery or a higher CRP level during the first episode of the disease as well as for those with a history of coronary disease or diabetes mellitus ($P = 0.005$, $P = 0.030$, $P = 0.009$, $P = 0.049$, and $P = 0.025$, respectively), (Table 4).

Fifteen patients (21.4%) died from the CDI group and 7 (9.6%) from the control group ($P = 0.050$). The median number of days between CDI diagnosis and death was

**Table 1 Demographics and preoperative data.**

| Variable | Patients with CDI (n = 70) | Patients without CDI (n = 73) | P-value |
|---|---|---|---|
| Age (years) | 73.0 [64.0–78.0] | 67.0 [58.0–74.0] | **0.005** (M) |
| Male, n(%) | 45 (64.3) | 51 (69.9) | 0.595 |
| BMI (kg/m²) | 28.0 [24.4–32.0] | 27.2 [24.4–31.3] | 0.557 |
| BMI > 25 kg/m², n(%) | 50 (71.4) | 47 (64.4) | 0.431 |
| Logistic EuroSCORE (points) | 6.0 [5.0–9.0] | 5.0 [4.0–7.0] | **<0.001** (M) |
| LVEF (%) | 55.0 [41.2–60.0] | 50.0 [40.0–60.0] | 0.144 |
| LVEF < 40%, n(%) | 14 (20.0) | 14 (19.2) | 0.901 |
| Hospitalization < 6 month before the surgery, n(%) | 62 (88.6) | 46 (63.0) | **0.001** (C) |
| Comorbidities, n(%) | | | |
|     Atherosclerosis | 33 (47.1) | 21 (28.8) | **0.036** (C) |
|     Dyslipidemia | 57 (81.4) | 50 (68.5) | 0.112 |
|     Coronary disease | 49 (70.0) | 35 (47.9) | **0.012** (C) |
|     Previous myocardial infarction | 21 (30.0) | 18 (24.7) | 0.597 |
|     Atrial fibrillation | 29 (41.4) | 22 (30.1) | 0.217 |
|     Diabetes mellitus | 28 (40.0) | 19 (26.0) | 0.110 |
|     History of malignant neoplasms | 9 (12.9) | 2 (2.7) | 0.050 |
|     Chronic kidney disease | 20 (28.6) | 14 (19.2) | 0.262 |
|     Thyroid disease | 16 (22.9) | 12 (16.4) | 0.450 |
|     Peptic ulcer disease | 15 (21.4) | 7 (9.6) | 0.084 |
|     COPD | 7 (10.0) | 8 (11.0) | 1.000 |
| Medicines, n(%) | | | |
|     Statins | 59 (84.3) | 47 (64.4) | **0.012** (C) |
|     Acetylsalicylic acid | 60 (85.7) | 44 (60.3) | **0.001** (C) |
|     Beta blockers | 60 (85.7) | 46 (63.0) | **0.004** (C) |
|     ACE inhibitors | 42 (60.0) | 32 (43.8) | 0.077 |
|     Insulin | 18 (25.7) | 10 (13.7) | 0.110 |
|     PPI | 61 (87.1) | 49 (67.1) | **0.008** (C) |
|     Corticosteroids | 4 (5.7) | 4 (5.5) | 1.000 |
| Laboratory parameters | | | |
|     WBC (×10³/μL) | 7.4 [6.3–8.8] | 8.2 [6.7–11.0] | **0.007** (M) |
|     WBC > 10,000 μL, n(%) | 8 (11.4) | 24 (32.9) | **0.002** (C) |
|     Platelets (×10³/μL) | 200.5 [159.2–261.5] | 216.0 [169.0–266.0] | 0.241 |
|     Platelets < 140,000 μL, n(%) | 9 (12.9) | 5 (6.8) | 0.227 |
|     Hematocrit (%) | 39.3 [35.6–42.7] | 38.5 [34.0–41.6] | 0.368 |
|     Hemoglobin (g/dL) | 13.0 [11.9–14.2] | 12.5 [11.0–14.0] | 0.241 |
|     Hemoglobin < 10 g/dL, n(%) | 9 (12.9) | 12 (16.4) | 0.545 |
|     RBC (×10⁶/μL) | 4.4 [4.0–4.9] | 4.4 [3.8–4.7] | 0.283 |
|     Creatinine (μmol/L) | 91.5 [82.2–115.8] | 99.0 [77.0–115.0] | 0.928 |
|     Creatinine > 100 μmol/L, n(%) | 26 (37.1) | 32 (43.8) | 0.415 |

**Notes.**

Continuous variables are presented as median (interquartile range). Categorical variables are presented as number (percentage).

(C), Pearson's chi-square test; (M), Mann–Whitney U test; ACE, angiotensin-converting enzyme; BMI, body mass index; COPD, chronic obstructive pulmonary disease; LVEF, left ventricular ejection fraction; PPI, proton pump inhibitors; RBC, red blood cells; WBC, white blood cells.

**Table 2  Intraoperative and postoperative data.**

| Variable | Patients with CDI ($n = 70$) | Patients without CDI ($n = 73$) | P-value |
|---|---|---|---|
| Type of surgery, $n$(%) | | | 0.239 |
|     CABG | 25 (35.7) | 21 (28.8) | 0.374 |
|     VHS | 29 (41.4) | 34 (46.6) | 0.535 |
|     CABG+VHS | 7 (10.0) | 6 (8.2) | 0.711 |
|     Aortic surgery | 6 (8.6) | 12 (16.4) | 0.156 |
|     CABG+aortic surgery | 3 (4.3) | 0 (0.0) | 0.115 |
| Reoperation, $n$(%) | 5 (7.1) | 7 (9.6) | 0.821 |
| Timing of surgery, $n$(%) | | | 0.217 |
|     Elective | 41 (58.6) | 51 (69.9) | |
|     Emergent | 29 (41.4) | 22 (30.1) | |
| Additional antibiotic, $n$(%) | 46 (65.7) | 32 (43.8) | **0.014** (C) |
| LVEF after surgery (%) | 45.0 [40.0-50.0] | 45.0 [35.0-55.0] | 0.839 |
| LVEF < 30%, $n$(%) | 5 (7.1) | 10 (13.7) | 0.201 |
| Aortic cross-clamp time (min) | 65.0 [36.2–89.5] | 69.0 [49.0–92.0] | 0.517 |
| CPB time (min) | 104.5 [74.0–154.5] | 125.0 [85.0–165.0] | 0.156 |
| Postoperative drainage (ml/first 24 h) | 530.0 [332.5–940.0] | 520.0 [380.0–810.0] | 0.981 |
| Postoperative drainage > 1,000 ml/first 24 h, $n$(%) | 17 (24.3) | 12 (16.4) | 0.243 |
| Inotropic agents, $n$(%) | 48 (68.6) | 53 (72.6) | 0.730 |
| IABP after surgery, $n$(%) | 2 (2.9) | 2 (2.7) | 1.000 |
| Accompanying infections, $n$(%) | | | |
|     Wound infection | 13 (18.6) | 13 (17.8) | 1.000 |
|     VAC therapy | 11 (15.7) | 5 (6.8) | 0.157 |
|     Positive blood cultures | 10 (14.3) | 15 (20.5) | 0.444 |
| Transfusion, $n$(%) | | | |
|     Red blood cells ≥2 units | 43 (61.4) | 41 (56.2) | 0.639 |
|     Plasma ≥2 units | 23 (32.9) | 23 (31.5) | 1.000 |
|     Platelets ≥1 unit | 21 (30.0) | 24 (32.9) | 0.849 |
| Laboratory parameters | | | |
|     WBC ($\times 10^3/\mu$L) | 9.7 [7.0–13.6] | 12.9 [9.3–15.6] | **0.007** (M) |
|     WBC > 13,000 $\mu$L, $n$(%) | 21 (30.0) | 36 (49.3) | **0.018** (C) |
|     Platelets ($\times 10^3/\mu$L) | 122.5 [86.5–154.0] | 130.0 [93.0–168.0] | 0.261 |
|     Platelets < 100,000 $\mu$L, $n$(%) | 23 (32.9) | 22 (30.1) | 0.726 |
|     Hematocrit (%) | 28.6 [26.8–30.9] | 29.4 [27.0–32.4] | 0.104 |
|     Hemoglobin (g/dL) | 9.6 [8.8–10.5] | 9.8 [9.0–10.7] | 0.229 |
|     Hemoglobin < 8.0 g/dL, n(%) | 9 (12.9) | 6 (8.2) | 0.366 |
|     RBC ($\times 10^6/\mu$L) | 3.2 [3.0–3.4] | 3.3 [3.0–3.5] | 0.118 |
|     Plasma glucose (mmol/L) | 11.5 [9.7–12.8] | 9.2 [8.1–10.7] | **<0.001** (M) |
|     Stress hyperglycemia, $n$(%) | 48 (68.6) | 24 (32.9) | **<0.001** (C) |
| In-hospital death, $n$(%) | 15 (21.4) | 7 (9.6) | **0.050** (C) |

**Notes.**

Continuous variables are presented as median (interquartile range). Categorical variables are presented as number (percentage).

(C), Pearson's chi-square test; (M), Mann–Whitney $U$ test; CABG, coronary artery bypass grafting; CPB, cardiopulmonary bypass; IABP, intra-aortic balloon pump; LVEF, left ventricular ejection fraction; RBC, red blood cells; VHS, valvular heart surgery; VAC, Vacuum-assisted closure; WBC, white blood cells.

**Table 3** Course of *Clostridioides difficile* infection general data.

| Variable | Patients with CDI (*n* = 70) |
|---|---|
| Antibiotic before the first episode od CDI, *n*(%)[a] | |
|     Cefazolin | 62 (88.6) |
|     Ceftriaxone | 27 (38.6) |
|     Fluoroquinolone | 15 (21.4) |
|     Vancomycin | 10 (14.3) |
|     Amoxicillin/Clavulanic acid | 8 (11.4) |
|     Piperacillin/Tazobactam | 8 (11.4) |
|     Meropenem | 7 (10.0) |
|     Clindamycin | 6 (8.6) |
|     Rifampicin | 4 (5.7) |
|     Gentamicin | 3 (4.3) |
|     Cloxacillin | 3 (4.3) |
|     Teicoplanin | 3 (4.3) |
|     Cefuroxime | 2 (2.9) |
|     Ceftazidime | 1 (1.4) |
|     Sulfamethoxazole/Trimethoprim | 1 (1.4) |
|     Erythromycin | 1 (1.4) |
|     Metronidazole | 1 (1.4) |
|     Colistin | 1 (1.4) |
| Severity of CDI, *n*(%) | |
|     Non-severe | 31 (44.3) |
|     Severe | 34 (48.6) |
|     Fulminant | 5 (7.1) |
| Times of the CDI diagnosis (days) | |
|     Between hospital admission and CDI diagnosis | 12.0 [6.2–30.5] |
|     Between the surgery and CDI diagnosis | 9.0 [5.0–27.2] |
|     Length of ICU stay before the CDI diagnosis | 4.0 [2.0–7.0] |
|     Assisted ventilation before the CDI diagnosis | 1.0 [1.0–2.0] |
|     Time of hospitalization after CDI diagnosis | 11 [6.2–20.0] |
| Treatment, *n*(%) | |
|     None | 0 (0.0) |
|     Metronidazole | 30 (42.8) |
|     Vancomycin | 4 (5.7) |
|     Both | 36 (51.4) |
|     Fidaxomicin | 0 (0.0) |

**Notes.**

Continuous variables are presented as median (interquartile range). Categorical variables are presented as number (percentage).

[a]Some patients used more than one antibiotic, therefore the percentage sum does not equal 100%.

CDI, *Clostridioides difficile* infection; ICU, intensive care unit.

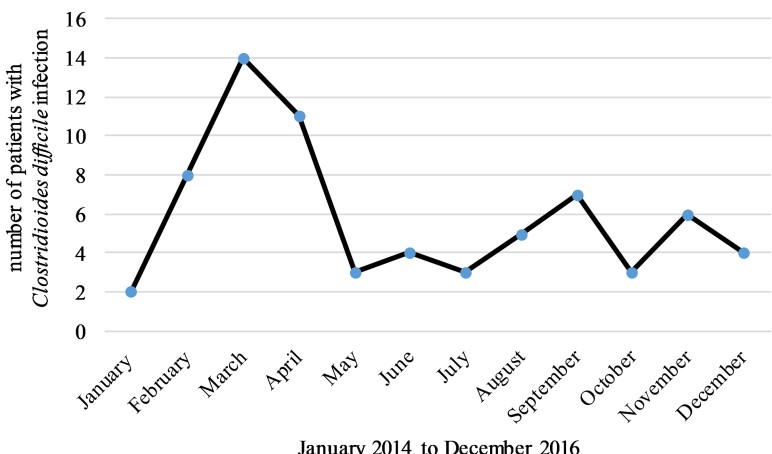

January 2014 to December 2016

**Figure 1** **Most cases of CDI occurred in March and the least number of cases were found in January.**

14.0 [4.0; 25.0]. Emergent procedures, prolonged stay in the intensive care unit, longer mechanical ventilation and high WBC count during the diarrhea were associated with higher mortality in patients with CDI ($P = 0.05$, $P = 0.041$, $P = 0.004$ and $P = 0.007$, respectively), (Table 4).

## DISCUSSION

The study showed that cardiac surgery related factors such as a type and timing of surgery, aortic cross-clamp and cardiopulmonary bypass time, volume of postoperative drainage, administration of blood products and value of postoperative ejection fraction were not correlated with the risk of CDI.

Interestingly, in our study, patients with CDI had a lower level of WBC count both before and after surgery in comparison to the control group. It is known that WBC, especially neutrophils, play an important role in the immune response against CD toxin A (*Kelly et al., 1994*), therefore, patients with a low WBC count have a higher risk of acquiring CDI (*Gorschlüter et al., 2001*). Unfortunately, in our study, neutrophil level was not determined, only overall WBC count was analyzed.

In contrast to other studies, our results did not show that diabetes mellitus was a risk factor for CDI (*Furuya-Kanamori et al., 2015a*). However, we did demonstrate that patients with high glucose levels and stress hyperglycemia during the early postoperative period were at greater risk for development of CDI. This finding is consistent with the results of a study by Gelijns et al., who demonstrated the association of hyperglycemia with CDI (*Gelijns et al., 2014*). Similarly, Kirkwood et al. showed that postoperative hyperglycemia was associated with an increased risk of CDI (*Kirkwood et al., 2018*). Therefore, prevention and treatment of hyperglycemia after cardiac surgery should be taken into consideration.

In our study, other risk factors for CDI were similar to those from non-cardiac surgery reports (Belto, Litofsky & Humphries, 2019). Our findings confirm the results of many studies that older age is an independent risk factor for CDI (*McFarland, Surawicz & Stamm,*

**Table 4  Univariate analysis of the *Clostridioides difficile* infection group stratified by relapse and in-hospital death.**

| Variable | Non-relapse group (*n* = 61) | Relapse group (*n* = 9) | *P*-value | Discharged alive (*n* = 55) | In-hospital death (*n* = 15) | *P*-value |
|---|---|---|---|---|---|---|
| Age (years) | 73.0 [64.0–78.0] | 72.0 [63.0–78.0] | 0.562 | 73.0 [64.0–78.0] | 76.0 [63.0–80.0] | 0.456 |
| Male, *n*(%) | 41 (67.2) | 4 (44.4) | 0.265 | 37 (67.3) | 8 (53.3) | 0.487 |
| BMI (kg/m$^2$) | 27.8 [24.1–30.6] | 35.1 [29.4–37.9] | **0.005** (M) | 28.3 [25.8–31.8] | 24.3 [21.3–32.1] | 0.174 |
| BMI > 25 kg/m$^2$, *n*(%) | 41 (67.2) | 9 (100.0) | 0.052 | 43 (78.2) | 7 (46.7) | **0.025** (C) |
| Logistic EuroSCORE (points) | 7.0 [5.0–9.0] | 5.0 [3.0–6.0] | 0.061 | 7.0 [5.0–9.0] | 6.0 [5.0–7.5] | 0.812 |
| LVEF baseline (%) | 55.0 [40.0–60.0] | 50.0 [47.0–65.0] | 0.552 | 55.0 [48.5–60.0] | 50.0 [32.5–60.0] | 0.290 |
| LVEF baseline < 40%, *n*(%) | 13 (21.3) | 1 (11.1) | 0.675 | 9 (16.4) | 5 (33.3) | 0.161 |
| Hospitalization < 6 month before the surgery, *n*(%) | 53 (86.9) | 9 (100) | 0.584 | 53 (96.4) | 9 (60.0) | **0.001** (C) |
| Comorbidities, *n*(%) | | | | | | |
|   Atherosclerosis | 30 (49.2) | 3 (33.3) | 0.485 | 29 (52.7) | 4 (26.7) | 0.133 |
|   Dyslipidemia | 49 (80.3) | 8 (88.9) | 1.000 | 49 (89.1) | 8 (53.3) | 0.004 |
|   Coronary disease | 40 (65.6) | 9 (100.0) | **0.049** (C) | 41 (74.5) | 8 (53.3) | 0.125 |
|   Previous myocardial infarction | 16 (26.2) | 5 (55.6) | 0.116 | 15 (27.3) | 6 (40.0) | 0.356 |
|   Atrial fibrillation | 27 (44.3) | 2 (22.2) | 0.289 | 23 (41.8) | 6 (40.0) | 1.000 |
|   Diabetes mellitus | 21 (34.4) | 7 (77.8) | *0.025* | 24 (43.6) | 4 (26.7) | 0.372 |
|   History of malignant neoplasms | 9 (14.8) | 0 (0.0) | 0.592 | 8 (14.5) | 1 (6.7) | 0.672 |
|   Chronic kidney disease | 19 (31.1) | 1 (11.1) | 0.430 | 17 (30.9) | 3 (20.0) | 0.528 |
|   Thyroid disease | 14 (23.0) | 2 (22.2) | 1.000 | 13 (23.6) | 3 (20.0) | 1.000 |
|   Peptic ulcer disease | 14 (23.0) | 1 (11.1) | 0.672 | 13 (23.6) | 2 (13.3) | 0.497 |
|   COPD | 6 (9.8) | 1 (11.1) | 1.000 | 6 (10.9) | 1 (6.7) | 1.000 |
| Medicines, *n*(%) | | | | | | |
|   Statins | 50 (82.0) | 9 (100.0) | 0.336 | 50 (90.9) | 9 (60.0) | 0.009 |
|   Acetylsalicylic acid | 52 (85.2) | 8 (88.9) | 1.000 | 51 (92.7) | 9 (60.0) | 0.005 |
|   Beta blockers | 51 (83.6) | 9 (100.0) | 0.339 | 48 (87.3) | 12 (80.0) | 0.437 |
|   ACE inhibitors | 37 (60.7) | 5 (55.6) | 1.000 | 34 (61.8) | 8 (53.3) | 0.766 |
|   Insulin | 15 (24.6) | 3 (33.3) | 0.685 | 15 (27.3) | 3 (20.0) | 0.744 |
|   PPI | 53 (86.9) | 8 (88.9) | 1.000 | 49 (89.1) | 12 (80.0) | 0.392 |
|   Corticosteroids | 4 (6.6) | 0 (0.0) | 1.000 | 3 (5.5) | 1 (6.7) | 1.000 |
| Laboratory parameters | | | | | | |
|   WBC baseline (×10$^3$/μL) | 7.4 [6.2–8.7] | 7.9 [6.6–8.8] | 0.605 | 7.5 [6.3–8.8] | 7.4 [6.1–8.4] | 0.626 |
|   WBC baseline > 10,000 μL, *n*(%) | 8 (13.1) | 0 (0.0) | 0.584 | 8 (14.5) | 0 (0.0) | 0.187 |
|   WBC postoperative (×10$^3$/μL) | 9.7 [6.8–13.4] | 9.6 [9.5–15.6] | 0.273 | 9.6 [6.9–13.4] | 10.2 [7.4–14.4] | 0.836 |
|   WBC postoperative > 13,000 μL, *n*(%) | 17 (27.9) | 4 (44.4) | 0.437 | 16 (29.1) | 5 (33.3) | 0.758 |
|   WBC during first CDI (×10$^3$/μL) | 13.8 [8.9–17.9] | 8.3 [6.4–15.1] | 0.129 | 12.4 [8.3–15.8] | 18.6 [9.4–37.8] | **0.007** (M) |

**Table 4** (*continued*)

| Variable | Non-relapse group (*n* = 61) | Relapse group (*n* = 9) | *P*-value | Discharged alive (*n* = 55) | In-hospital death (*n* = 15) | *P*-value |
|---|---|---|---|---|---|---|
| WBC during first CDI > 15,000, *n*(%) | 26 (42.6) | 3 (33.3) | 0.726 | 19 (34.5) | 10 (66.7) | **0.025** (C) |
| Plasma glucose postoperative (mmol/L) | 11.5 [9.1–12.5] | 13.0 [10.7–15.0] | **0.030** (M) | 11.5 [9.1–12.8] | 11.5 [10.2–12.8] | 0.637 |
| Stress hyperglycemia postoperative, *n*(%) | 40 (65.6) | 8 (88.9) | 0.255 | 37 (67.3) | 11 (73.3) | 0.761 |
| CRP during first CDI (mg/L) | 55.9 [29.0–113.0] | 169.0 [74.0–444.0] | **0.009** (M) | 71.0 [32.0–141.0] | 80.0 [33.0–233.5] | 0.587 |
| CRP during first CDI > 60 mg/L, *n*(%) | 26 (49.1) | 9 (100.0) | **0.004** (C) | 28 (50.90) | 7 (46.7) | 0.742 |
| Type of surgery, *n*(%) | | | 0.716 | | | 0.955 |
| CABG | 22 (36.1) | 3 (33.3) | | 20 (36.4) | 5 (33.3) | |
| VHS | 25 (41.0) | 4 (44.4) | | 22 (40.0) | 7 (46.7) | |
| CABG+VHS | 6 (9.8) | 1 (11.1) | | 6 (10.9) | 1 (6.7) | |
| Aortic surgery | 6 (9.8) | 0 (0.0) | | 5 (9.1) | 1 (6.7) | |
| CABG+aortic surgery | 2 (3.3) | 1 (11.1) | | 2 (3.6) | 1 (6.7) | |
| Reoperation, *n*(%) | 5 (8.2) | 0 (0.0) | 1.000 | 3 (5.5) | 2 (13.3) | 0.290 |
| Timing of surgery, *n*(%) | | | 0.071 | | | **0.050** (C) |
| Elective | 33 (54.1) | 8 (88.9) | | 36 (65.5) | 5 (33.3) | |
| Emergent | 28 (45.9) | 1 (11.1) | | 19 (34.5) | 10 (66.7) | |
| Additional antibiotic, *n*(%) | 45 (73.8) | 1 (11.1) | 0.001 | 36 (65.5) | 10 (66.7) | 1.000 |
| LVEF after surgery (%) | 45.0 [40.0–50.0] | 48.0 [40.0–55.0] | 0.797 | 45.0 [40.0–50.0] | 40.0 [32.5–57.5] | 0.603 |
| LVEF after surgery < 30%, *n*(%) | 5 (8.2) | 0 (0.0) | 1.000 | 3 (5.4) | 2 (13.3) | 0.290 |
| Aortic cross-clamp time (min) | 66.0 [36.0–92.0] | 59.0 [40.0–83.0] | 0.958 | 67.0 [38.0–89.0] | 43.0 [34.5–95.5] | 0.596 |
| CPB time (min) | 105.0 [66.0–160.0] | 90.0 [81.0–114.0] | 0.806 | 105.0 [82.5–154.0] | 100.0 [65.5–185.0] | 0.983 |
| Postoperative drainage (ml/first 24 h) | 510.0 [340.0–950.0] | 570.0 [280.0–700.0] | 0.752 | 510.0 [345.0–865.0] | 550.0 [315.0–1435.0] | 0.699 |
| Postoperative drainage > 1,000 ml/first 24 h, *n*(%) | 15 (24.6) | 2 (22.2) | 1.000 | 12 (21.8) | 5 (33.3) | 0.497 |
| Inotropic agents, *n*(%) | 42 (68.9) | 6 (66.7) | 1.000 | 37 (67.3) | 11 (73.3) | 0.761 |
| IABP after surgery, *n*(%) | 2 (3.3) | 0 (0.0) | 1.000 | 2 (3.6) | 0 (0.0) | 1.000 |
| Accompanying infections, *n*(%) | | | | | | |
| Wound infection | 11 (18.0) | 2 (22.2) | 0.670 | 9 (16.4) | 4 (26.7) | 0.455 |
| VAC therapy | 9 (14.8) | 2 (22.2) | 0.625 | 8 (14.5) | 3 (20.0) | 0.691 |
| Positive blood cultures | 9 (14.8) | 1 (11.1) | 1.000 | 4 (7.3) | 6 (40.0) | **0.005** (C) |
| Transfusion, *n*(%) | | | | | | |
| Red blood cells ≥2 units | 39 (63.9) | 4 (44.4) | 0.292 | 33 (60.0) | 10 (66.7) | 0.864 |
| Plasma ≥2 units | 21 (34.4) | 2 (22.2) | 0.708 | 16 (29.1) | 7 (46.7) | 0.226 |
| Platelets ≥1 unit | 18 (29.5) | 3 (33.3) | 1.000 | 15 (27.3) | 6 (40.0) | 0.356 |

| Variable | Non-relapse group ($n = 61$) | Relapse group ($n = 9$) | P-value | Discharged alive ($n = 55$) | In-hospital death ($n = 15$) | P-value |
|---|---|---|---|---|---|---|
| Times of the CDI diagnosis (days) | | | | | | |
| Length of ICU stay before the CDI diagnosis | 4.0 [2.0–7.0] | 3.0 [2.0–6.0] | 0.958 | 3.0 [2.0–5.5] | 6.0 [2.5–20.0] | **0.041** (M) |
| Assisted ventilation before the CDI diagnosis | 1.0 [1.0–3.0] | 1.0 [1.0–1.0] | 0.108 | 1.0 [1.0–1.5] | 3.0 [1.0–14.5] | **0.004** (M) |
| Severity of CDI, $n$(%) | | | 0.865 | | | 0.182 |
| Non-severe | 26 (42.6) | 5 (55.6) | | 27 (49.1) | 4 (26.7) | |
| Severe | 30 (49.2) | 4 (44.4) | | 25 (45.5) | 9 (60.0) | |
| Fulminant | 5 (8.2) | 0 (0.0) | | 3 (5.5) | 2 (13.3) | |

**Notes.**

Continuous variables are presented as median (interquartile range). Categorical variables are presented as number (percentage).

(C), Pearson's chi-square test; (M), Mann–Whitney $U$ test; ACE, angiotensin-converting enzyme; BMI, body mass index; CABG, coronary artery bypass grafting; CDI, *Clostridioides difficile* infection; COPD, chronic obstructive pulmonary disease; CPB, cardiopulmonary bypass; CRP, C-reactive protein; IABP, intra-aortic balloon pump; LVEF, left ventricular ejection fraction; PPI, proton pump inhibitors; WBC, white blood cells; ICU, intensive care unit; VHS, valvular heart surgery; VAC, Vacuum-assisted closure.

*1990*; *De Roo & Regenbogen, 2020*; *Flagg et al., 2014*; *Belton, Litofsky & Humphries, 2019*). Unlike other reports, we did not find a correlation between female gender and development of CDI (*Flagg et al., 2014*; *Ge et al., 2018*).

Some studies suggest an association between PPI and CDI risk whereas others do not confirm this correlation (*McFarland, Surawicz & Stamm, 1990*; *Eze et al., 2017*; *Furuya-Kanamori et al., 2015a*; *Faleck et al., 2016*; *Maes, Fixen & Linnebur, 2017*). The exact mechanism of proliferation of CD in patients using PPI remains unclear (*Maes, Fixen & Linnebur, 2017*). In our study, patients with CDI significantly more often used PPI. The necessity of PPI use should be carefully evaluated for hospital patients, especially those already receiving antibiotics.

Patients with CDI more frequently have chronic illnesses (*Eze et al., 2017*; *De Roo & Regenbogen, 2020*; *Furuya-Kanamori et al., 2015a*; *Kirkwood et al., 2018*). In our study atherosclerosis, ischemic heart disease and history of malignant neoplasms were correlated with CDI. Furthermore, patients with CDI significantly more often received statins, b-blockers and acetylsalicylic acid. This correlation could be explained by the fact that these drugs are used to treat the comorbidities that are associated with development of CDI. The correlation between EuroSCORE and CDI could be explained similarly.

Time of hospital stay is an important risk factor for the development of CDI (*Flagg et al., 2014*; *Chalmers et al., 2016*). Spores can remain in the hospital environment for several months and are difficult to remove with traditional disinfectants (*Barbut, 2015*). We also showed the significant role of hospitalization time in the risk of CDI.

Besides periprocedural antimicrobial prophylaxis, some patients received an additional antibiotic due to accompanying infections, and these patients had greater chance of contracting CDI. In our study cefazolin, ceftriaxone and fluoroquinolone were the most frequently used antibiotics (Table 3). The relationship between antibiotic treatment and the risk of CDI has also been demonstrated in other studies (*McFarland, Surawicz & Stamm, 1990*; *Eze et al., 2017*; *De Roo & Regenbogen, 2020*; *Furuya-Kanamori et al., 2015a*; *Kirkwood et al., 2018*; *Ge et al., 2018*; *Kazakova et al., 2020*; *Jachowicz et al., 2020*). It should

be remembered that any antibiotic may be the cause of CDI, even those used to treat CDI (*McDonald et al., 2018*). Appropriate antibiotic management can help to reduce the risk of postoperative CDI.

In this study, a large number of patients received metronidazole for CDI treatment (Table 3). Our analysis was performed between 2014 and 2016 and treatment of CDI was consistent with the 2010 recommendations (*Cohen et al., 2010*). Current guidelines confirm that metronidazole has a lower efficacy compared with vancomycin and they support the use of vancomycin over metronidazole in CDI (*McDonald et al., 2018*).

In our study the most cases of CDI occurred in March. This result is comparable with other studies that have shown that CDI has a similar seasonal pattern characterized by a peak in spring and lower frequencies in summer and autumn months (*Furuya-Kanamori et al., 2015b*). *Jachowicz et al. (2020)* showed an additional peak of healthcare associated CDI occurring from October to December. The mechanisms responsible for the seasonality of CDI remain poorly understood, although it has been proposed that the observed seasonality is associated with a higher incidence of respiratory infections, which leads to an intensified use of antibiotics during winter and spring months.

One of the most challenging aspects for patients with CDI is the recurrence of the disease after successful initial therapy is completed, which has been observed in between 15 to 35% of patients (*De Roo & Regenbogen, 2020*; *Dharbhamulla et al., 2019*). We observed a relapse in about 13% of patients with CDI. It is possible that this discrepancy was the result of different times of observation. In our study we assessed relapses that occurred only during the same hospitalization stay as the surgery. The causes of recurrent CDI are also unclear (*Dharbhamulla et al., 2019*). In our study, the risk of recurrent CDI significantly increased for patients with diabetes mellitus or ischemic heart disease and for those with higher BMI or higher glucose level on the day of surgery. Similar to Predrag et al., we also revealed an association of relapses with high CRP during the first episode of CDI (*Predrag et al., 2020*).

Data on mortality for patients who acquired CDI after surgery varies from 2.5 to 27.7% and is much higher than mean in-hospital death after cardiac surgery (1–4%) (*Vondran et al., 2018*; *Flagg et al., 2014*; *Mazzeffi et al., 2014*). Our result of a 21% mortality rate validate these findings. We showed that patients that had emergent procedures, prolonged stay in the intensive care unit, longer mechanical ventilation or a high WBC count during an episode of diarrhea have a higher risk of death. These findings are consistent with other results (*Ricciardi et al., 2007*; *Furuya-Kanamori et al., 2015a*; *Furuya-Kanamori et al., 2015b*; *Gelijns et al., 2014*; *Vondran et al., 2018*; *Flagg et al., 2014*; *Debast, Bauer & Kuijper, 2014*). In our study, prolonged hospitalization time before surgery was a risk factor for CDI but correlated with lower mortality. Longer hospital stay could help to ensure that patients enter elective surgery in the best condition possible in contrast to emergent procedures.

## Limitations

This study has several limitations. It was a retrospective study based on available data obtained during patients' cardiac procedure hospital stay. Therefore, the true incidence of CDI could be higher due to a lack of information regarding potential post-discharge diagnosis of the disease. The size of the study group was limited and patients were very

heterogeneous. It is probable that analysis of patients with one type of cardiac surgery may provide different results.

## CONCLUSION

In conclusion, this study did not reveal any specific cardiac surgery-related risk factors for development of CDI. Also, frequency of relapse and mortality rate were similar to non-cardiac surgery. Studies in larger cohorts are needed to confirm these findings.

### Funding

This article was supported by science found on John Paul II Hospital, Krakow, Poland (no. FN7/2020 to D.P.). The funders had no role in study design, data collection and analysis, decision to publish, or preparation of the manuscript.

### Grant Disclosures

The following grant information was disclosed by the authors:
John Paul II Hospital, Krakow, Poland: FN7/2020.

### Competing Interests

The authors declare there are no competing interests.

### Author Contributions

- Anna Rzucidło-Hymczak, Hubert Hymczak and Dariusz Plicner conceived and designed the experiments, performed the experiments, analyzed the data, prepared figures and/or tables, authored or reviewed drafts of the paper, and approved the final draft.
- Aldona Olechowska-Jarząb, Anna Gorczyca, Boguslaw Kapelak and Rafał Drwiła performed the experiments, authored or reviewed drafts of the paper, and approved the final draft.

### Human Ethics

The following information was supplied relating to ethical approvals (i.e., approving body and any reference numbers):

The local Research Ethics Committee of Andrzej Frycz Modrzewski Krakow University, Krakow, Poland approved this research (10/2019).

### Data Availability

The raw data are available in the Supplementary File.

### Supplemental Information

Supplemental information for this article can be found online at http://dx.doi.org/10.7717/peerj.9972#supplemental-information.

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
