# Peer review of "Clostridioides difficile infection after cardiac surgery: Assessment of prevalence, risk factors and clinical outcomes—retrospective study"

_PeerJ, doi:10.7717/peerj.9972_

## Round 0.1 · original submission · Minor Revisions

Although both reviewers have agreed for minor revisions, it is absolutely paramount that you follow and revise your manuscript scrupulously.

·

Basic reporting

the paper is written clearly and convey a clear and well-structured message

Experimental design

no comment

Validity of the findings

204/5000
the paper analyzes an interesting link between a key population such as patients undergoing cardiac surgery
Indeed cardiac diseases such as diabetes are major risk factors for CDI

Additional comments

The paper is remarkable because showed Clostridium infection complication in a peculiar surgical key population as cardiac surgery
The link between seasonal and antibiotic use in CDI was reported by other authors and there re evidence in the geographical area of Poland (Jachowicz, E.; Wałaszek, M.; Sulimka, G.; Maciejczak, A.; Zieńczuk, W.; Kołodziej, D.; Karaś, J.; Pobiega, M.; Wójkowska-Mach, J. Long-Term Antibiotic Prophylaxis in Urology and High Incidence of Clostridioides difficile Infections in Surgical Adult Patients. Microorganisms 2020, 8, 810.)
Please comment these findings take in account the use of third-generation cephalosporins and Fluoroquinolones reported in table 3.
Globally, guidelines and papers confirm that metronidazole has a lower efficacy compared with vancomycin and support the use of vancomycin over metronidazole in C difficile infection. To optimize vancomycin treatment evaluated in recurrent C difficile infection, it might be possible to use a pulsed or tapered regimen of vancomycin.( Guery Benoit, Galperine Tatiana, Barbut Frédéric. Clostridioides difficile: diagnosis and treatments BMJ 2019; 366 :l4609)
Please, comment why the treatment with metronidazole is over vancomycin in the paper

·

Basic reporting

The article is clear and unambiguous.
The article include sufficient introduction and background.
The references were appropriately cited

Experimental design

Original primary research within Aims and Scope of the journal.
The submission clearly define the research question.
The investigation has been conducted rigorously and to a high technical standard.
The research must has been conducted in conformity with the prevailing ethical standards in the field.
Methods are described with sufficient information to be reproducible by another investigator.

Validity of the findings

The statistical results should be improved.
The conclusions are appropriately stated,.

Additional comments

The study evaluated the course of CDI in patients after cardiac surgery.

Major concern
In this study additional statistical analyses should be performed to improvement the quality of the paper.

Comments
1) To facilitate the reading of the Tables with statistical tests, the authors should write in bold the significant p-values and in brackets the type of test used (for example (C) in the case of chi-square test)
2) Lines 112-113: “There was no correlation between gender and incidence of CDI (P = 0.595)”
The authors describe only some no significant results, but they do not describe all no significant results. Please, check it
3) Lines 118: the authors claim to consider all tests with p <0.05 as significant, but the history of malignant neoplasms has a p=0.05. Please, check it.
4) I suggest that authors use additional tests for parameters where a normal range is defined (Table 1, 2, and 4). In particular they should compare the percentages of patients with abnormal values (for example for BMI, Platelets, WBC, etc) . These tests could provide important information that may not emerge from the comparison between medians or means
5) In Table 2 the authors should also compare the subcategories of Type of surgery.
6) Line 128: “of cases were found in January (n=2, 3%), (P = 0.038 between March and rest of the months).
Which test is used by the authors? If a multiple-comparison chi-square test was used, which post hoc test was used to find the most frequent proportion?

---

## Round 0.2 · accepted · Accept

The authors addressed properly the comments of the reviewers.